# In Vitro Characterization of the Human Skeletal Stem Cell-like Properties of Primary Bone-Derived Mesenchymal Stem/Stromal Cells in Patients with Late and Early Hip Osteoarthritis

**DOI:** 10.3390/life12060899

**Published:** 2022-06-15

**Authors:** Lara Jasenc, Klemen Stražar, Anže Mihelič, Rene Mihalič, Rihard Trebše, Gregor Haring, Matjaž Jeras, Janja Zupan

**Affiliations:** 1Department of Clinical Biochemistry, Faculty of Pharmacy, University of Ljubljana, Askerceva 7, 1000 Ljubljana, Slovenia; lj8200@student.uni-lj.si; 2Department of Orthopaedic Surgery, University Medical Centre Ljubljana, Zaloska 9, 1000 Ljubljana, Slovenia; klemen.strazar@kclj.si; 3Faculty of Medicine, University of Ljubljana Vrazov trg 2, 1000 Ljubljana, Slovenia; rihard.trebse@ob-valdoltra.si; 4Valdoltra Orthopaedic Hospital, Jadranska 31, 6280 Ankaran, Slovenia; anze.mihelic@ob-valdoltra.si (A.M.); rene.mihalic@ob-valdoltra.si (R.M.); 5Institute of Forensic Medicine, Faculty of Medicine, University of Ljubljana., Korytkova 2, 1000 Ljubljana, Slovenia; gregor.haring@mf.uni-lj.si; 6Celica, Biomedical Center, d.o.o., Tehnoloski Park 24, 1000 Ljubljana, Slovenia

**Keywords:** human skeletal stem cells (hSSCs), late osteoarthritis (OA), early OA, bone-derived mesenchymal stem/stromal cells (MSCs), podoplanin (PDPN), CD73, CD164, CD146, trilineage differentiation, immunophenotyping

## Abstract

Human skeletal stem cells (hSSCs) were recently identified as podoplanin (PDPN)/CD73/CD164-positive and CD146-negative cells that decline with age, and play a role in the pathogenesis of osteoarthritis (OA). The aim of this study was to identify the hSSC-like properties of bone-derived mesenchymal stem/stromal cells (MSCs) of patients with late and early OA. **Methods**: First, we performed gene expression profiling for the hSSC markers in 32 patients with late and early OA, and donors without OA. Having identified the low expression of hSSC markers in late OA patients, we further performed trilineage differentiation and immunophenotyping for hSSC makers in the selected subsets from each donor group. **Results**: Our results show no differences in osteogenesis, chondrogenesis, and adipogenesis between the MSCs from the three groups. However, the immunophenotyping shows lower CD164 in MSCs from early OA patients in comparison with late and no OA subjects (*p* = 0.002 and *p* = 0.017). **Conclusions**: Our study shows that the in vitro hSSC-like properties of bone-derived MSCs are similar in patients with early and late OA, and in donors without OA. However, the lower percentage of CD164-positive MSCs in early OA patients indicates the potential of CD164 as a marker of the onset of OA.

## 1. Introduction

Osteoarthritis (OA) is a degenerative disorder primarily of the joints. However, the process also affects other tissues of the musculoskeletal system and, therefore, causes impaired mobility. The incidence of OA increases with age, and since life expectancy is prolonging, the global prevalence of hip and knee OA is approaching 5% [1,2]. The condition can be classified into primary and secondary OA, depending on the cause of disease. While the cause is unclear in primary OA, secondary OA can occur as a consequence of an endocrine system disorder, pathological anatomy, post-traumatic OA, and inflammatory arthritis [3]. Injuries such as anterior cruciate ligament and meniscal tears cause post-traumatic knee OA, particularly in young people whose joints are otherwise normal without the coexisting pathomechanics and bone shape alterations, due to an overload of reduced cartilage contact areas and/or shear stresses resulting from instability [4]. One of the major independent risk factors is aging associated with changes in the musculoskeletal system [4,5]. Slowly developing chronic joint pain is the most frequent clinical symptom accompanied by stiffness, joint instability, and deformations [6]. Studies suggest that most people with painful OA already have extensive structural disease including malalignment, which may preclude successful stabilization, or reversal, of the disease [1]. Due to the lack of non-pharmacological and pharmacological treatment of OA, surgical interventions are currently the most effective in treating OA.

Mesenchymal stem/stromal cells (MSCs), with a capacity to regenerate connective tissues, present an appealing option for the treatment of the damaged joints [2,3,4]. MSCs are resident cells of the numerous connective tissues that are able to self-renew, and exhibit multilineage differentiation to bone, cartilage, stroma, muscle, and fat. Although the mechanisms by which MSCs exert their regenerative effects are not fully understood, it is generally accepted that immunomodulatory and anti-inflammatory properties on the one hand, and repair and restoration mechanisms on the other hand, have a synergistic, or additive, effect on the osteoarthritic joint [7].

In current clinical practice, the most frequently used tissue sources of MSCs for joint regeneration are autologous bone-marrow and adipose tissue [2,3,4,5]. However, recent studies show that MSCs derived from patients with degenerative disorders, such as OA, might have impaired capabilities for tissue regeneration [6,7,8,9]. Also, MSC exhaustion and a decrease in their regenerative potential was suggested as a hallmark of aging [10]. Another hurdle associated with the more efficient use of these cellular therapies is the absence of reliable cell-surface markers to identify the MSCs with regenerative properties. Recently, significant progress was made with the identification of the human skeletal stem cells (hSSCs), and their potential as cellular therapies for OA [11,12]. Chan et al. demonstrate that hSSCs, identified by podoplanin (PDPN)^+^CD146^−^CD73^+^CD164^+^ immunophenotype, generate the progenitors of bone, cartilage, and stroma, but not fat [12]. These self-renewing cells are present in various skeletal tissues, and can undergo local expansion in response to acute skeletal injury [12]. Moreover, the same group of scientists further demonstrate that aging is associated with progressive loss of hSSCs and their diminished chondrogenesis in the joints, hence, contributing to the onset of OA [11]. However, following microfracture surgery, and localized co-delivery of bone morphogenetic protein 2 (BMP2) and soluble vascular endothelial growth factor receptor antagonist (sVEGFR1), resident hSSCs can be induced to generate cartilage for treatment of localized chondral disease in OA [11].

Following these recent findings, the aim of the present study was to identify the hSSC-like properties of bone-derived MSCs of patients with late and early OA. First, we performed gene expression profiling for hSSC markers in patients with late and early OA, and donors without OA. Having identified the cluster of samples from late OA patients with a lower expression of the hSSC markers, we further sought to identify if the bone-derived primary cells exert different trilineage potential and hSSC immunophenotype in vitro. 

## 2. Methods

### 2.1. Donor Inclusion, Tissue Harvesting, and Primary Cell Isolation

The samples from patients with late stage OA of the hip and post mortem donors without hip OA were used from our previous study [13]. The late OA patients were included during routine total hip arthroplasty at the Valdoltra Orthopaedic Hospital, Ankaran, Slovenia. Donors without OA were included during routine autopsies at the Institute of Forensic Medicine, Faculty of Medicine, University of Ljubljana, Slovenia. The patients with early OA were included during hip arthroscopy procedures at the Department of Orthopaedic Surgery, University Medical Centre Ljubljana, Slovenia. The stage of the OA was diagnosed by clinical examination and plain X-rays, according to Tönnis [14]. The causes of the OA in the group of late (advanced) OA patients were most commonly primary OA, hip dysplasia, and cam impingement. The early stage of OA was diagnosed preoperatively by clinical examination, plain radiographs, and magnetic resonance arthrography, and was further confirmed during arthroscopy, i.e., classified as grade I, according to Tönnis [14]. The causes of the OA in the group of early OA patients included labrum lesion, degenerative labrum changes with ruptures, and pincer impingement. In post mortem donors, the photographs of the exposed femoral heads were taken, and the absence of hip degeneration was confirmed via macroscopic examination of the hip. The exclusion criteria for all donors included history of inflammatory arthritis, metastatic cancer, and disorders that affect bone. From 2020, only the donors tested negative for SARS-CoV-2 were included. Approval for this study was obtained from the National Medical Ethics Committee of the Republic of Slovenia (reference numbers: 0120-523/2016/11, 0120-523/2016-2, and 0120-499/2020/7). Written informed consent to participate in this study was obtained from all patients, prior to inclusion in the study.

Trabecular bone tissue (approximately 1 cm^3^ in size) was harvested from femoral head of each subject. All of the tissues harvested were stored in low-glucose Dulbecco’s Modified Eagle’s medium (DMEM; Biowest), supplemented with 10% fetal bovine serum (Gibco), 1% glutamine, and 2% penicillin and streptomycin (all Biowest), until cell isolation at the Faculty of Pharmacy, University of Ljubljana, Slovenia.

The primary cell isolation from bone biopsies was performed, as described previously [6,13]. Briefly, bone pieces were washed thoroughly in phosphate-buffered saline, and incubated at 37 °C in 1 mg/ mL collagenase solution (Roche) for 3 h. The resulting suspensions of tissue and cells were passed through a 70 µm cell strainer (Corning). Aliquots of freshly isolated cells were seeded using StemMACS MSC expansion media kit XF, human (Miltenyi Biotec), supplemented with 1% glutamine, and 2% penicillin and streptomycin (all Biowest). The cells were incubated at 37 °C under 5% humidified CO_2_. The cells after p0 were routinely seeded at 5000 cells/cm^2^ in low-glucose DMEM (Biowest), supplemented with 10% fetal bovine serum (Gibco), 1% glutamine, and 2% penicillin and streptomycin (all Biowest), until enough cells were obtained for the planned analyses. The study design and the analyses are summarized in Figure 1.

### 2.2. RNA Isolation, Reverse Transcription, and Gene Expression Profiling

Culture-expanded cells (between passage p1 and p3) were used for RNA isolation. For late OA patients and no OA donors, the RNA and cDNA samples were used from our previous study [13]. For patients with early OA, total RNA was extracted using qGOLD Total RNA kits (VWR), and the cDNA was synthesized using high-capacity cDNA Reverse Transcription kits (Thermo Fisher Scientific).

Gene expression analysis was performed according to MIQE guidelines [15], and as described previously [6,13,16]. Briefly, quantitative polymerase chain reaction (qPCR) was performed using 5× HOT FIREPol EvaGreen qPCR Supermix (Solis BioDyne), according to manufacturer protocol. The sequences of the primers (Macrogen, Sigma-Aldrich) used to measure genes encoding hSSC markers and osteogenesis- and adipogenesis-related genes were used from our [17], and other, previous studies [18,19,20,21]. All qPCR experiments were performed in triplicates, using a LightCycler 480 II (Roche). Gene expression data was obtained using standard curve. All of the data were normalized to glyceraldehyde-3-phosphate dehydrogenase (*GAPDH*).

### 2.3. Trilineage Differentiation

Trilineage differentiation was performed, as described previously [6,13,16,17]. Briefly, for osteogenesis and adipogenesis, the cells were seeded as four technical replicates. Two replicates were used for histological assessment, and two replicates for RNA isolation and gene expression analysis. The treated replicates received either osteogenic medium (growth medium supplemented with 5 mM β-glycerophosphate, 100 nM dexamethasone, and 50 mg/mL ascorbic acid-2-phosphate [all Sigma]), or adipogenic medium (growth medium supplemented with 500 nM dexamethasone, 10 µM indomethacin, 50 µM iso-butyl-methyl-xanthine, and 10 µg/mL insulin [all Sigma]). The controls received growth medium without the adipogenic or osteogenic supplements. After 21 days, the osteogenic cultures were stained with 2% alizarin red S, and the adipogenic cultures with oil red O (both Sigma-Aldrich, St. Louis, MI, USA). After staining, the cells were imaged using Evos XL (Life Technologies, Carlsbad, CA, USA). The bound-alizarin red S was extracted using 5% SDS in 0.5 M hydrochloric acid (both Sigma), and quantified at 405 nm using Safire 2 microplate reader (Tecan, Männedorf, Switzerland). The osteogenic potential was calculated as the concentration of alizarin red S (mM). The adipogenic potential was calculated as the numbers of oil-red-O-positive adipocytes per number of seeded cells, using the ImageJ software [22]. For chondrogenesis, cell pellets were formed as duplicates of 150,000 cells, suspended in chondrogenic medium (high-glucose DMEM [Biowest, Nuaillé, France], 100 nM dexamethasone [Sigma], 1% insulin–transferrin–selenium [Sigma], 50 mg/mL ascorbic acid-2-phosphate [Sigma], and 1% penicillin/streptomycin [Biowest]). The treated pellets received 10 ng/mL transforming growth factor ß1 (TGF-ß1; Thermo Fisher Scientific), and the controls received medium without TGF-ß1. After 21 days, the pellets were fixed in 10% neutral-buffered formalin (Sigma), and processed for paraffin sections at the Institute of Pathology, Faculty of Medicine, University of Ljubljana. The 5 µm paraffin sections were stained with toluidine blue (Sigma) and von Kossa histology, imaged using Evos XL (Life Technologies), and analyzed according to the Bern score [23]. Immunofluorescence for collagen type II (Col2A1), as described previously [6,13,16], was also performed. Briefly, goat anti-Col2 antibody, conjugated with Alexa Fluor 488 (Southern Biotech Cat#1320-30) was used in 1:50 dilution with DAPI Prolong Gold Antifade (Thermo Fisher Scientific, Waltham, MA, USA). The sections were imaged and inspected for the presence of Col2A1 using Evos FL (Life Technologies).

### 2.4. Immunophenotyping

Immunophenotyping was performed using flow cytometry, as described previously [6,13,16,17]. Culture-expanded cells between p1–p5 were immunophenotyped using the following antibodies: anti-CD73 (clone AD2, Miltenyi Biotec, Bergisch Gladbach, Germany), anti-CD164 (clone 67D2), anti-PDPN (clone NC-08), anti-CD146 (clone P1H12), anti-CD45 (clone 2D1), and anti-CD235a clone (clone HI264) (all BioLegend, San Diego, CA, USA). The fixable viability dye eFluor 780 (Thermo Fisher Scientific) was used to determine cell viability. Data were acquired on an Attune NxT instrument (Thermo Fisher Scientific, Waltham, MA, USA), and analyzed using FlowJo v10.7.2. software.

### 2.5. Statistical Analysis

The Shapiro–Wilk test was used to test the normality of the distributions of the data. To compare age and body mass index (BMI) between the patients with late and early OA, and the donors without OA, a two-way ANOVA, with Bonferroni corrections for multiple testing, was used. Since age proved to be statistically significantly different between the tested groups of subjects, age was used as a covariate when testing the differences between the three groups of donors, using the general linear model (GLM) and a Bonferroni post hoc. To compare the categorical data (male/female ratio, data on positive von Kossa, and Col2A1 staining) between the three donor groups, a chi-squared test was used. The statistical analyses were performed using IBM SPSS Statistics version 27, and Graph Pad Prism version 8.4.3 for Windows (GraphPad Software, San Diego, CA, USA, www.graphpad.com, accessed on 10 May 2022). *p* values < 0.05 were considered as statistically significant. A heat map was generated, as described previously [16,17] using the online Heatmapper software [24].

## 3. Results

### 3.1. Study Subjects

In the first stage of the study, when the expression profiling for hSSC marker genes is carried out, we include 32 subjects. The basic characteristics of the subjects per donor group are given in Table 1. Significant differences are obtained for age between late and early OA (**** *p* < 0.0001), and late and no OA (*p* = 0.003; one-way ANOVA, with Bonferroni multiple comparison tests).

In the second stage of the study, when in vitro analyses of the primary cells is performed, we include 15 subjects. The basic characteristics of the subjects per donor group are given in Table 2. The range of age for the subjects in late OA group is 74 to 88 years; in the early OA group, 45 to 54 years; and in the no OA group, 28 to 76 years. Significant differences are obtained for age between late and early OA (** *p* = 0.004), and late and no OA (* *p* = 0.014; one-way ANOVA, with Bonferroni multiple comparison tests).

### 3.2. Gene Expression Profiling Identified Clusters of Samples from Late OA Patients with Low Expression of hSSC Markers

The expression profiling of genes recently identified as markers of hSSCs (i.e., *PDPN*, *CD73*, *CD164*, and *CD146*) [11,12] was carried out on 32 samples of cDNA derived from the in vitro-cultured primary cells between p1 and p3, used for the current and previous studies [13]. Hierarchical clustering of hSSC genes identifies two clusters with a high expression of negative marker *CD146*, and low expression of positive markers *CD73*, *PDPN,* and *CD164*, as illustrated in the heat map in Figure 2a (blue circles). These mainly encompass samples of primary cells from late OA patients (3/5 and 4/4 samples). 

For the comparison of the expression profiles of hSSC gene markers between the three donor groups (Figure 2b), no significant differences are seen (*p* > 0.05; general linear model with age as covariate, and a Bonferroni post hoc comparison).

### 3.3. The Primary Bone-Derived Cells Show Similar Osteogenic Potential between the Tested Groups of Donors 

To evaluate the differences in the osteogenic potential of the primary bone-derived cells in vitro between the three donor groups, alizarin red S staining and quantification are used, and gene expression of specific markers of osteogenesis measured (Figure 3). A comparison of the alizarin red S concentrations between the donor groups does not show any differences (Figure 3a and Appendix A). Even though the primary cells from late OA are less osteogenic in comparison with early OA and no OA groups of donors (mean 3.5 mM for late OA, 4.0 mM for early OA, and 5.0 mM for no OA), the results do not reach the statistical significance (*p* = 0.380; general linear model with age as covariate, and a Bonferroni post hoc comparison). Similarly, gene expression measurement of specific markers of osteogenesis (Figure 3b), namely osteocalcin (*OC*), collagen type I (*COL1A1*), and alkaline phosphatase (*ALP*), do not show any differences between the donor groups (*p* = 0.899 for *OC*, *p* = 0.373 for *COL1A1,* and *p* = 0.380 for *ALP*; general linear model with age as covariate, and a Bonferroni post hoc comparison).

### 3.4. No Differences in Chondrogenic Potential of the Primary Bone-Derived Cells between the Tested Groups of Donors 

To evaluate the differences in the chondrogenic potential of the primary bone-derived cells in vitro between the three donor groups, toluidine blue staining and evaluation according to Bern are used (Figure 4a,b and Appendix A). A comparison of the Bern scores between the donor groups does not show any differences (Figure 4a). Even though the primary cells from late OA are less chondrogenic in comparison with early OA and no OA groups of donors (mean 3 for late OA, 6.75 for early OA, and 6.40 for no OA), the results do not reach the statistical significance (*p* = 0.252; general linear model with age as covariate, and a Bonferroni post hoc comparison). To assess the rate of mineralization in chondrogenic pellets, von Kossa histology is performed (Figure 4c and Appendix A). Apart from one sample in the late OA group, and one in the without OA group, that show slightly weak positive staining (Appendix A), no staining, indicating calcium mineralization, is observed (Figure 4c). No statistically significant difference is observed between the three donor groups (*p* > 0.05; chi-squared test). To assess the rate of hyaline cartilage formation, the immunofluorescence for Col2A1 is performed (Figure 4d). The positive staining for Col2A1 is not observed in any of the samples in late OA group; slight staining is observed in three out of four samples in early OA group; and two samples out of five in the no OA group show intensive Col2A1 staining. No statistically significant difference is observed between the three donor groups (*p* > 0.05; chi-squared test).

### 3.5. Similar Adipogenic Potential of the Primary Bone-Derived Cells between the Tested Groups of Donors 

To evaluate the differences in the adipogenic potential of the primary bone-derived cells in vitro between the three donor groups, quantification of the oil-red-O-positive adipocytes is performed, and gene expression of specific markers of adipogenesis measured (Figure 5). A comparison of the % of the oil-red-O-positive adipocytes between the donor groups does not show any differences (Figure 5a). The primary bone-derived cells from all three donor groups show a similar tendency for the adipogenesis (mean 0.55% for late OA, 0.42% for early OA, and 0.51% for no OA, (*p* = 0.613; general linear model with age as covariate, and a Bonferroni post hoc comparison). Gene expression measurement of specific markers of adipogenesis (Figure 5b), namely, adiponectin (*ADIPOQ*), peroxisome proliferator-activated receptor γ (*PPARG*), and fatty acid-binding protein 4 (*FABP4*) show significant differences for *FABP4* between late and early OA (* *p* = 0.026), and early and no OA (* *p* = 0.012; general linear model with age as covariate, and a Bonferroni post hoc comparison). Adipogenic cells from early OA patients show the lowest expression of *FABP4* in comparison with late and no OA donors (mean 31.781 for late OA, 0.590 for early OA, and 374.422 for no OA).

### 3.6. The Primary Bone-Derived Cells from Early OA Patients Show Lower Percentage of the Positive Marker CD164 in Comparison with the Late OA Patients and No OA Donors

To evaluate the differences in the percentage of the hSSC markers expressed on the primary bone-derived cells in vitro between the three donor groups, immunophenotyping is performed (Figure 6). A comparison of the % of the hSSC markers shows statistically significant differences in the expression of the positive marker CD164 between all three group of donors (Figure 6a). The primary bone-derived cells from early OA patients show the lowest percentage of CD164-positive cells (0.47%), in comparison with those from late OA patients (3.04%, ** *p* = 0.002), and with those without OA (3.12%, * *p* = 0.017). Also, primary bone-derived cells from early OA donors show a significantly lower percentage of the negative marker, CD146, in comparison with no OA donors (* *p* = 0.038).

## 4. Discussion

Mesenchymal stem/stromal cells (MSCs) present a promising option for regenerative treatment of various degenerative disorders, and, in particular, for joint degeneration in OA [2,4,5]. This debilitating disorder of the joints is on the rise with the aging of the world population [25,26], with the effective treatment options currently limited to only major surgical interventions. The cellular therapies most commonly used in current clinical practice are derived from bone marrow and adipose tissue. However, the regenerative capacities of the MSCs in these cellular therapies are dependent on the age and the pathophysiological condition of the donor, and, in particular, on the concomitant presence of already established OA [6,7,8,10,11,13]. Due to the lack of the standard markers to identify the MSCs with the regenerative capacities, the search for more effective MSCs therapies is still on the rise.

With the recent discovery of the subpopulation of MSCs in humans, namely hSSCs, new research directions emerged towards better characterization of MSCs to be used in regenerative medicine, or as biomarkers of early tissue degeneration. Natively resident in skeletal tissues, the cells with the immunophenotype PDPN^+^CD146^−^CD73^+^CD164^+^ respond to local skeletal injury, and regenerate cartilage and bone [12]. The number of hSSCs declines with age, and in degenerative disorders, such as OA [11]. However, with the combination of microfracture, and codelivery of BMP2 and sVEGFR1, the endogenous hSSCs can be stimulated to regenerate articular cartilage, and heal local chondral lesions in OA [11].

Based on these, and our recent findings on the exhaustion of the MSCs in patients with primary OA of the hip [6,13], we aimed to investigate if the hSSC-like properties of the bone-derived MSCs differ between the patients with late and early OA of the hip. Post mortem donors with no degenerative changes of the hip (no OA) were used as controls, similar to our previous study [13].

In the present study, we first tested if the bone-derived MSCs of the three groups of donors, i.e., patients with late and early OA, and donors without OA, differ in the gene expression of the hSSC markers. Having identified two clusters of samples from late OA patients with a low expression profile for hSSC (Figure 2a), we further tested the subsets of bone-derived MSCs from the three donor groups for their in vitro properties, such as trilineage differentiation and hSSC immunophenotype. Due to the differences in age between the tested donor groups (Table 1 and Table 2), and the well-recognized evidence on the influence of age on regenerative properties of MSCs [10], we included age as a covariate in our statistical analyses. After the correction for age, no significant differences were observed for any of the trilineage potential. Even though the osteogenic and chondrogenic potentials are the lowest in the group of bone-derived MSCs from late OA patients, the results do not reach statistical significance, due to the age correction (Figure 3 and Figure 4). The difference in osteogenic potential between late and early OA patients suggest the exhaustion of the bone-derived MSCs in late OA, as observed in our previous study [13]. However, other authors find reduced chondrogenic and adipogenic potential of bone-marrow MSCs in late OA patients, while their osteogenic potential is similar to healthy donors [8]. However, the difference in osteogenic potential between late and early OA patients in our study is not statistically significant, hence, the verification using larger group of samples is needed. As for the adipogenesis, the MSCs from all three donor groups show similar adipogenic potential in vitro (Figure 5). The only statistically significant difference is observed for the lower expression of the adipogenesis-related gene *FABP4* encoding fatty acid-binding protein 4 gene in early OA patients, in comparison with late and no OA donors (Figure 5b). Since Chan et al. show that hSSCs in vivo are able to form cartilage, bone, and stroma, but not fat, it might be that other MSCs subpopulations present in our in vitro cultures of bone-derived MSCs are sources of adipogenic cells.

Comparing the immunophenotype of the in vitro-expanded bone-derived MSCs between the three donor groups, significant differences are obtained for the positive marker CD164, and the negative marker CD146 (Figure 6). Also known as endolyn, CD164 is a member of sialomucin family, previously associated with human hematopoietic progenitor–stromal cells [27]. In contrast to CD73, which has been a well-recognized positive marker for more than a decade [28], until the identification of hSSCs, CD164 was not used as marker of MSCs. Our finding on the low percentage of CD164-positive cells in bone-derived MSCs from early OA patients, in comparison with late OA patients and no OA donors, opens new possibilities on the utilization of CD164 as a cellular marker of the onset of OA of the hip. It might be that the lower amount of the CD164-positive, bone-derived MSCs in the trabecular bone of patients with early OA changes contributes to the onset of OA. To prove that, the immunophenotyping of the freshly isolated trabecular-bone-derived primary cells from early OA patients, and donors without OA, is required. The third positive marker of hSSC, PDPN, also known as gp38, emerged in previous studies as a marker of MSCs, in particular for synovium-derived MSCs [29,30], and also as a marker of fibroblasts [31]. A recent study also shows that PDPN regulates the migration of MSCs, as these cells upregulate PDPN at sites of infection, chronic inflammation, and cancer [32]. Interestingly, the negative marker of hSSC, CD146, which shows a lower percentage in the cells from early OA patients in comparison with no OA donors, was previously used as positive marker of MSCs [33]. Most commonly, CD146 was associated with MSCs of a perivascular origin, due to its function as key cell adhesion protein in vascular endothelial cell activity and angiogenesis [34,35].

There are also limitations to the present study. The major limitation is that it was not possible to perform all of the in vitro analyses for the complete study cohort; i.e., for all of the 32 donors included in the first stage of our study. Isolation and culture expansion of primary cells is a relatively long, and sometimes tedious, procedure, and our previous studies show that it can differ greatly between the donors [6,13,16]. In vitro analyses, in particular trilineage differentiation, require substantial numbers of cells. Some of the primary cells did not expand enough for the multiple analyses to be performed here. However, many of these limitations also apply to other studies, and the size of the present study cohort is comparable to other studies [7,8,36]. Nevertheless, the results of the in vitro analyses obtained in the second stage of our study are only preliminary, and need confirmation in larger group of subjects. On the other hand, the advantage of the present study is the comprehensive approach to the analysis of primary bone-derived MSCs with the inclusion of the control group, i.e., post mortem donors with no degenerative changes of the hip. Our, and other previous, studies show that MSCs from post mortem donors possess similar MSC-like properties as those from living donors [13,36]. Following the findings of the recent high-impact studies [11,12], the results of the current study contribute to the translation of this knowledge towards more effective cellular therapies, and understanding of the pathophysiology of the hip OA.

To summarize, the present study indicates that bone-derived MSCs from patients with late and early OA, and donors without OA, possess comparable potential for osteogenesis, chondrogenesis, and adipogenesis in vitro. There are also no differences in immunophenotype for the recently identified markers of hSSCs, except for the lower percentage of the CD164-positive and CD146-negative cells in early OA patients.

These data underpin the potential of CD164 as a cellular marker of the bone-derived MSCs at the early onset of hip OA. They also warrant further studies, in particular on bone-derived MSCs, to establish the contribution of CD164-positive MSCs to tissue regeneration, and their exact role in OA pathophysiology.

## 5. Conclusions

To summarize, the present study shows that primary bone-derived MSCs from patients with late and early OA, as well as donors with no OA of the hip, exert comparable in vitro properties, such as osteogenic, chondrogenic, and adipogenic potential, and immunophenotype for the recently identified hSSC markers, except for CD164 and CD146. Given that the cells from early OA patients show a lower expression of the positive marker CD164, these data suggest the usefulness of the CD164 as a marker of the bone-derived MSCs at the early onset of OA, and warrant further studies in these patient groups.

## Figures and Tables

**Figure 1 life-12-00899-f001:**
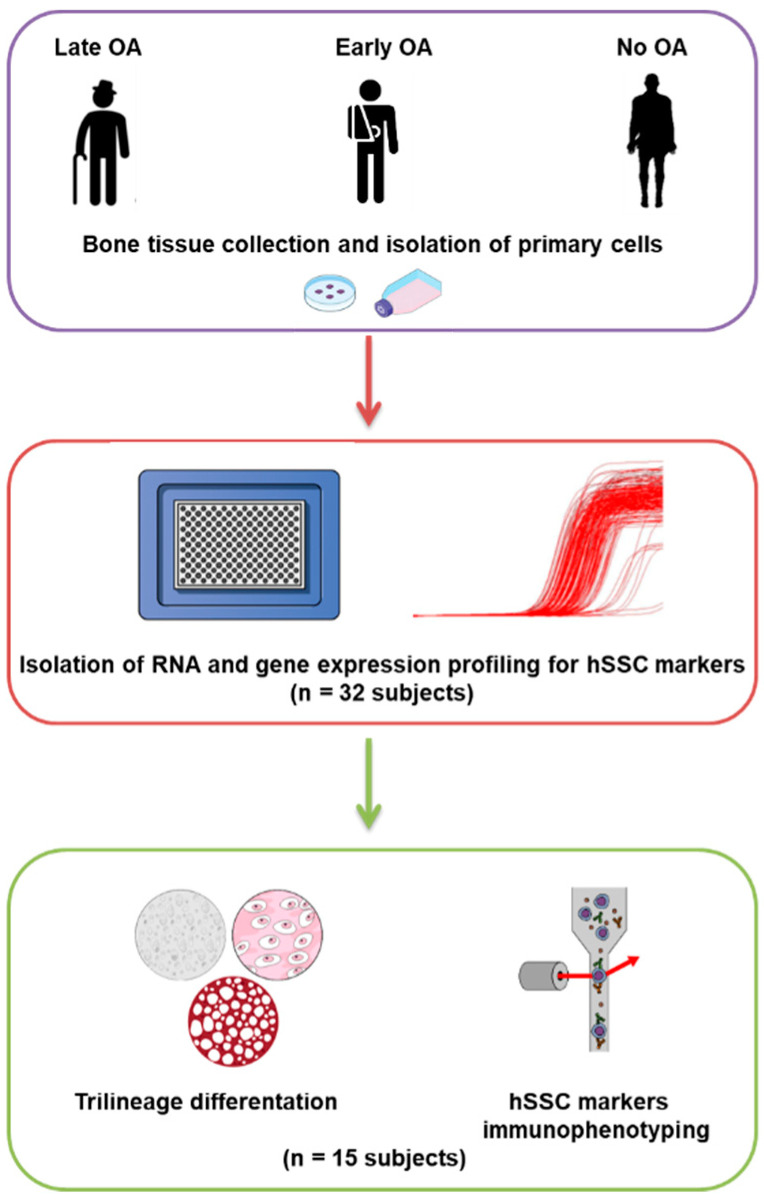
Study design, subject groups, and analyses. The subjects in our study were divided into 3 donor groups (violet circle), i.e., patients with late OA of the hip, patients with early OA of the hip, and post mortem donors with no degenerative changes of the hip (no OA group). All subjects had their trabecular bone from the femoral head harvested for primary cell isolation. The primary cells from subjects with late OA and without OA were used from our previous study [13]. The current study comprised of two stages. In the first stage (red circle), gene expression profiling for human skeletal stem cell (hSSC) markers was performed on cDNA samples from bone-derived MSCs from 32 subjects. Based on these results, we further selected 15 samples (5 samples per subject group) for the in vitro analyses, such as trilineage differentiation and immunophenotyping, for hSSC markers (green circle).

**Figure 2 life-12-00899-f002:**
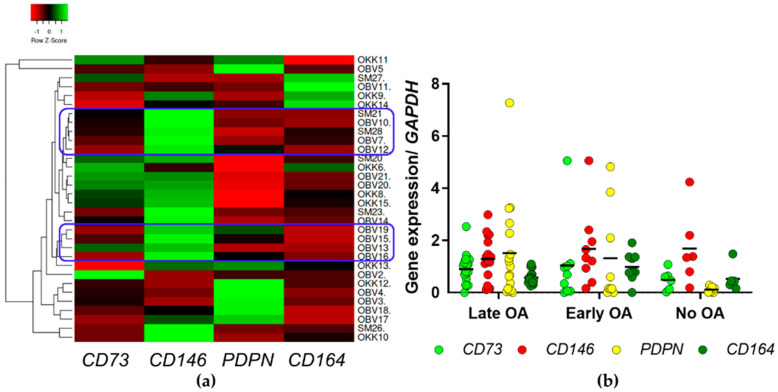
The results of the expression profiling for the hSSC marker genes. (**a**) Heat map analysis for hierarchical clustering of hSSC gene expression (columns, as indicated) in the primary cells from the three donor groups (rows). Green, gene expression higher than reference channel; red, gene expression lower than reference channel. Two clusters with high expression of the negative marker *CD146*, and low expression of the three positive markers are shown (right; boxed in blue), along with the clustering tree analysis (left). The majority of the samples in these two clusters are from late OA patients, i.e., 3 out of 5 in the upper cluster, and 4 out of 4 in the lower cluster. OBV, late OA samples; OKK, early OA samples; SM, samples with no OA. (**b**) Expression of the hSSC marker genes *CD73*, *CD146*, *PDPN*, and *CD164* for each donor group. Individual samples and means are shown. No significant differences in any of the measured gene are found (*p* > 0.05; general linear model with age as covariate, and a Bonferroni post hoc comparison). OA, osteoarthritis; *PDPN*, podoplanin.

**Figure 3 life-12-00899-f003:**
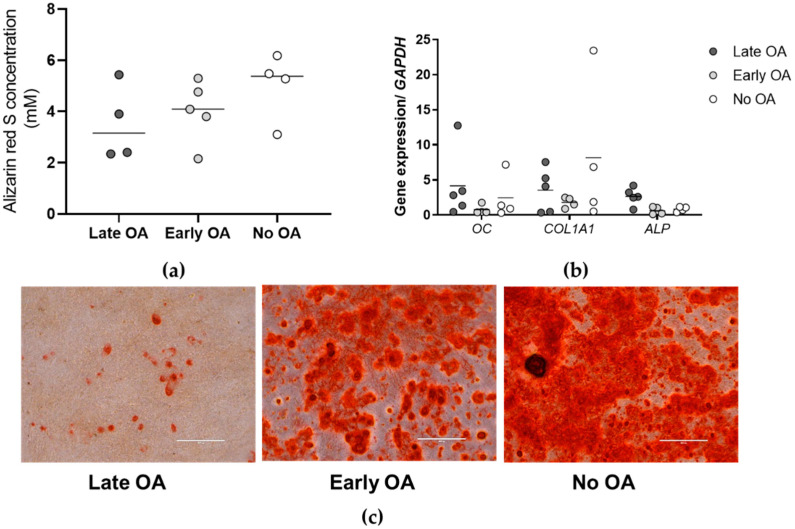
The results of the osteogenic potential of the primary bone-derived cells. (**a**) Alizarin red S concentration for each donor group is measured. Individual samples and means are shown. No significant differences are obtained between the three groups of donors (*p* = 0.380; general linear model with age as covariate, and a Bonferroni post hoc comparison). (**b**) Gene expression of specific markers of osteogenesis are measured. Individual samples and means are shown. Data are normalized to the reference gene glyceraldehyde-3-phosphate dehydrogenase (*GAPDH*). No significant differences are obtained between the three groups of donors, for any of the genes measured (*p* > 0.05; general linear model with age as covariate, and a Bonferroni post hoc comparison). (**c**) Representative images of the wells for each donor group (as indicated) stained with alizarin red S for rate of osteogenesis. Images for all donors are shown in Appendix A. Scale bars, 400 µm. OA, osteoarthritis; *OC*, osteocalcin; *COL1A1*, collagen type I; *ALP*, alkaline phosphatase.

**Figure 4 life-12-00899-f004:**
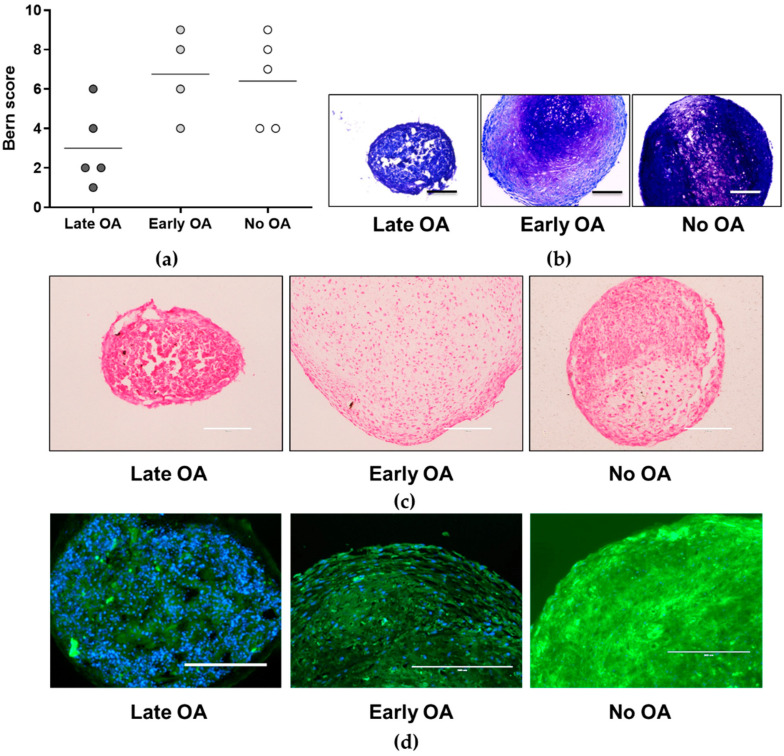
The results of the chondrogenic potential of the primary bone-derived cells. (**a**) Bern score of the toluidine blue-stained chondrogenic pellets for each donor group is determined. No significant differences are obtained between the three donor groups (*p* = 0.252; general linear model with age as covariate, and a Bonferroni post hoc comparison). Individual samples and means are shown. (**b**) Representative images of the toluidine blue-stained chondrogenic pellets for each group of donors are shown. Images for all donors are shown in Appendix A. (**c**) von Kossa histology shows no difference in the rate of mineralization between the three donor groups (*p* >0.05; chi-squared test). Representative images of the von Kossa-stained chondrogenic pellets for each group of donors are shown. Images for all donors are shown in Appendix A. (**d**) Immunofluorescence for the α-1 chain of type II collagen (Col2A1) shows no difference in the presence of the hyaline cartilage between the three donor groups (*p* > 0.05; chi-squared test). Representative images of the Col2A1-stained chondrogenic pellets for each group of donors are shown. Images for all donors are shown in Appendix A. Scale bars, 200 µm. OA, osteoarthritis.

**Figure 5 life-12-00899-f005:**
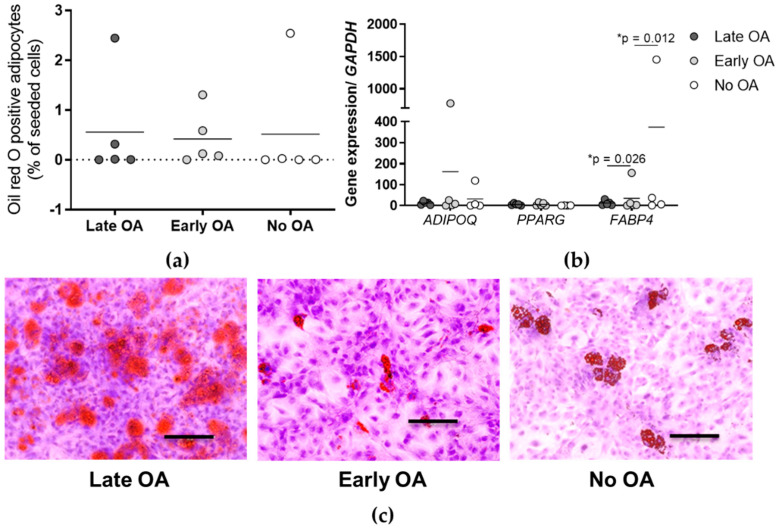
The results of the adipogenic potential of the primary bone-derived cells. (**a**) The percentage of the oil-red-O-positive adipocytes for each donor group is determined. Individual samples and means are shown. No significant differences are obtained between the three donor groups (*p* = 0.613; general linear model with age as covariate, and a Bonferroni post hoc comparison). (**b**) Gene expression of specific markers of adipogenesis is measured. Data are normalized to the reference gene glyceraldehyde-3-phosphate dehydrogenase (*GAPDH*). Significant differences are obtained between for fatty acid-binding protein 4 gene (*FABP4*) between late and early OA, and early and no OA, as indicated (* *p* < 0.05; general linear model with age as covariate, and a Bonferroni post hoc comparison). (**c**) Representative images of the oil-red-O-stained adipocytes for each group of donors are shown. Images for all donors are shown in Appendix A. Scale bars, 200 µm. OA, osteoarthritis; *ADIPOQ*, adiponectin; *PPARG*, peroxisome proliferator-activated receptor γ; *FABP4*, fatty acid-binding protein 4.

**Figure 6 life-12-00899-f006:**
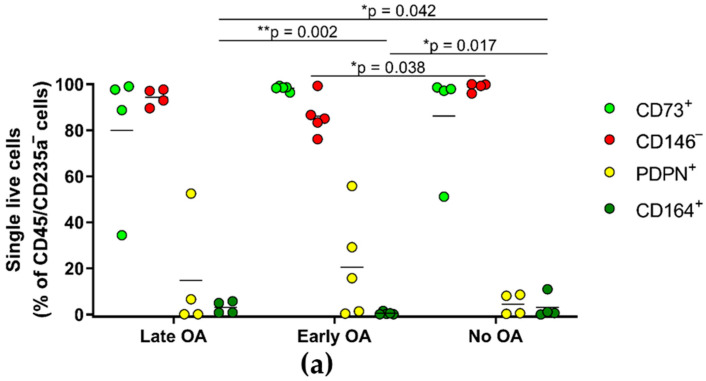
The results of the immunophenotyping for hSSC markers. (**a**) The percentage of the positive (CD73, PDPN, and CD164) and the negative (CD146) markers within single live CD45/CD235a-negative cells is determined. Significant differences are observed for the positive marker CD164 between all group of patients, and for the negative marker CD146 between early and no OA, as indicated (general linear model with age as covariate, and a Bonferroni post hoc comparison). * *p* < 0.05 and ** *p* < 0.01. (**b**) Representative dot plots for each hSSC marker in each group of donors are shown. OA, osteoarthritis; PDPN, podoplanin.

**Table 1 life-12-00899-t001:** Basic characteristics of the donors included for the expression profiling of hSSC marker genes.

Donor Group	N	Age (Years)	M/F	BMI (m/kg^2^)
Late OA	17	76 ± 11 ****/***	6/11	28.2 ± 4.6
Early OA	9	43 ± 14 ****	4/5	25.0 ± 2.7
No OA	6	53 ± 18 ***	3/3	28.2 ± 4.7

Shown are means ± standard deviations for age and body mass index (BMI). Significant differences are obtained for age between late and early OA (**** *p* <0.0001), and late and no OA (*** *p* = 0.003; one-way ANOVA, with Bonferroni multiple comparison tests).

**Table 2 life-12-00899-t002:** Basic characteristics of the donors included for the in-vitro analyses.

Donor Group	N	Age (Years)	M/F	BMI (m/kg^2^)
Late OA	5	80 ± 6 **/*	3/2	24.9 ± 2.5
Early OA	5	48 ± 4 **	1/4	25.6 ± 2.7
No OA	5	54 ± 20 *	2/3	27.1 ± 4.6

Shown are means ± standard deviation for age and body mass index (BMI). Significant differences are obtained for age between late and early OA (** *p* = 0.004), and late and no OA (* *p* = 0.014; one-way ANOVA, with Bonferroni multiple comparison tests).

## Data Availability

The data presented in this study are available on request from the corresponding author. The data are not publicly available due to restrictions, such as donor privacy protection, and ethical considerations.

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
