# Peer review of "In Vitro Characterization of the Human Skeletal Stem Cell-like Properties of Primary Bone-Derived Mesenchymal Stem/Stromal Cells in Patients with Late and Early Hip Osteoarthritis"

_life, 2022, doi:10.3390/life12060899_

Round 1

Reviewer 1 Report

Please find hereby my review of the article untitled “

In-vitro characterization of the human skeletal stem cell-like properties of primary bone-derived mesenchymal stem/ stromal cells in patients with late and early hip osteoarthritis” by Lara Jasenc et al. in Life

The submitted manuscript has a very interesting topic and a real clinical interest.

The present study indicates that bone-derived MSCs from patients with late and early OA and donors with no OA possess comparable potential for osteogenesis, chondrogenesis and adipogenesis in-vitro. There were also no differences in immunophenotype for the recently identified markers of hSSCs, except for the lower percentage of the CD164 positive and CD146 negative cells in early OA patients. 

The manuscript needs to be edited carefully (typographical errors…).

Authors must show results of histologic analyses for all groups, ex: Alizarin Red, Alcian blue et oil red O for the three axes of differentiation. For the pellet make also a Von Kossa !

Authors must show RT-PCR for specific markers of the differentiate cells ie col2, aggrecan, sox9, runx2, glut, PPARg, col X, col 1, OPN …..

Then make few immunohisto technique to confirm expression of Col2, FOXO ou OCN for example

To more discus the big difference of AR staining in bone differentiation (late OA versus No OA)

Reviewer 2 Report

Dear Authors,

Thank you for your manuscript. It was a pleasure to read as it is very interesting but does not exactly correspond to my area of expertise. So I have only a few minor technical comments.

On page 2, lines 88-93 text presents some study findings. It should be moved from the Introduction to the Results section.

Despite your study sample being described and published elsewhere, the reader might wonder for more information on study participants and their health status - duration and cause of the osteoarthritis in patients group, comorbid chronic conditions if this information is available. In section 3.1, please provide information on the age range in each study group (youngest-oldest). Also, in Tables 1 and 2, mean age and mean body mass index should be provided with the standard deviations. And for me, it is not clear which group is the reference when stars indicating different levels of statistical significance are provided for all three groups? In other words, ** p< 0.01 compared to what?

Wish the authors all the best in their further research.

Reviewer 3 Report

The manuscript by Zupan and coworkers deals with the characterization of hSSCs of osteoarthritis patients at early and late stage of disease. Although interesting, this research has been carried out on a small number of patients. In particular, the in vitro differentiation potential experiment has been performed on five samples per patient group. As the same Authors wrote, the isolation and primary cell expansion could vary significantly among donors (lines 384-386). Thus, a larger dataset is require to draw conclusions representative of each patient group.

Round 2

Reviewer 3 Report

In the revised manuscript, the Authors exhaustively discussed the limitations of their study, citing relevant literature to explain the rationale of their experimental design. Furthermore, the revised materials and methods and the results sections, have been significantly improved after the first round of revision.